# Knockout of the *OsNAC006* Transcription Factor Causes Drought and Heat Sensitivity in Rice

**DOI:** 10.3390/ijms21072288

**Published:** 2020-03-26

**Authors:** Bo Wang, Zhaohui Zhong, Xia Wang, Xiangyan Han, Deshui Yu, Chunguo Wang, Wenqin Song, Xuelian Zheng, Chengbin Chen, Yong Zhang

**Affiliations:** 1College of Life Sciences, Nankai University, Tianjin 300071, China; wangbo615@mail.nankai.edu.cn (B.W.); 2120160946@mail.nankai.edu.cn (X.W.); xy-han@mail.nankai.edu.cn (X.H.); deshui_yu@mail.nankai.edu.cn (D.Y.); wangcg@nankai.edu.cn (C.W.); songwq@nankai.edu.cn (W.S.); 2Department of Biotechnology, School of Life Sciences and Technology, Center for Informational Biology, University of Electronic Science and Technology of China, Chengdu 610054, China; zzh0409km@163.com (Z.Z.); zhengxl@uestc.edu.cn (X.Z.); 3Jiangsu Key Laboratory of Crop Genomics and Molecular Breeding, Agricultural College, Yangzhou University, Yangzhou 225009, China

**Keywords:** rice, NAC transcription factor, abiotic stresses, CRISPR-Cas9, transcriptome analysis

## Abstract

Rice (*Oryza sativa*) responds to various abiotic stresses during growth. Plant-specific NAM, ATAF1/2, and CUC2 (NAC) transcription factors (TFs) play an important role in controlling numerous vital growth and developmental processes. To date, 170 NAC TFs have been reported in rice, but their roles remain largely unknown. Herein, we discovered that the TF *OsNAC006* is constitutively expressed in rice, and regulated by H_2_O_2_, cold, heat, abscisic acid (ABA), indole-3-acetic acid (IAA), gibberellin (GA), NaCl, and polyethylene glycol (PEG) 6000 treatments. Furthermore, knockout of *OsNAC006* using the CRISPR-Cas9 system resulted in drought and heat sensitivity. RNA sequencing (RNA-seq) transcriptome analysis revealed that *OsNAC006* regulates the expression of genes mainly involved in response to stimuli, oxidoreductase activity, cofactor binding, and membrane-related pathways. Our findings elucidate the important role of *OsNAC006* in drought responses, and provide valuable information for genetic manipulation to enhance stress tolerance in future plant breeding programs.

## 1. Introduction

Rice (*Oryza sativa*) is one of the most important food crops for humans, and different abiotic stresses can affect plant growth and crop performance [1,2]. Salinity stress has a strong negative influence on plant growth [3,4]. Drought represents an extreme environment, and causes irreversible damage to rice growth and lowers crop yield and quality [5,6,7]. High temperature can impact rice flowering and can also reduce crop yield [8]. Plants have evolved various mechanisms to reduce the harmful effects of abiotic stresses [9], including regulating transcription factors (TFs) [10,11,12].

In rice, 2408 TFs have been identified and classified into 56 families, data provided by Plant Transcription Factor Database v3.0, Center for Bioinformatics, Peking University. Many TFs belonging to AP2/ERF (APETALA2/ethylene-responsive factor), bZIP (basic region/leucine zipper motif), NAC, MYB (v-myb avian myeloblastosis viral oncogene homolog) and WRKY families are believed to function in abiotic stress responses [11,13,14,15,16]. Among them, NAM, ATAF1/2, and CUC2 (NAC) TFs are a unique class in plants [17]. Many NAC TFs are involved in plant growth and development, and in responses to biotic and abiotic stresses [18,19]. Overexpressing *AtNAC07*, *AtNAC019*, and *AtNAC055* can enhance tolerance to drought in *Arabidopsis thaliana* [20]. Arabidopsis *ANAC092* (also known as *AtNAC2* or *ORE1*) is associated with the regulation of ethylene and hormone signaling, and overexpression can alter lateral root number, growth, and development [21]. Overexpression of the millet TF *OsNAC67* can increase rice tolerance to high salt and drought [22], while overexpression of *ZmSNAC1* can enhance the tolerance of maize to drought [23].

CRISPR/Cas9 gene editing technology is gradually applied to many genes related to rice breeding, which is of great significance for agricultural breeding [24]. The traditional transgenic technology is based on T-DNA insertion technology, and the vector transferred into the plant will not disappear [25]. CRISPR/Cas9-mediated genome editing has attracted people’s attention not only because of its simplicity, accuracy, and efficiency, but also because of its ability to produce non-transgenic plants [26]. The mutant plants that had produced the required mutations can lose the CRISPR/Cas9 vectors through several generations of character isolation. With the emergence of CRISPR/Cas9 gene editing technology, it is convenient for us to understand the gene function of plants. The new generation of breeding technology based on CRISPR/Cas9 editing system is gradually maturing.

In this work, we cloned the rice NAC TF-encoding gene *OsNAC006* (LOC_Os01g09550) and present evidence that mutations of this gene confer drought and heat sensitivity.

## 2. Results

### 2.1. Expression Profiling and Subcellular Localization of OsNAC006

We analyzed the expression profiles of eight representative tissues (root, stem, and leaf from seedlings, and root, stem, sheath, leaf, and panicle from the heading stage). RNA was extracted from different tissues and RT-qPCR was performed to determine the expression pattern of *OsNAC006*. The results indicated that *OsNAC006* was expressed in both seedling and heading stages in all tissues, with highest levels in stems and leaves (Figure 1A).

p*OsNAC006*::eGFP and eGFP (negative control) plasmids were infiltrated into rice protoplasts to examine the subcellular localization of *OsNAC006*. Confocal micrographs showed that the *OsNAC006*::eGFP fusion protein was localized to the nucleus, alongside the nuclear marker NLS::eGFP. Thus, the *OsNAC006* protein is localized to nuclei in cells (Figure 1B).

We also assessed whether and how *OsNAC006* contributes to the responses to abiotic stress and hormone treatment. *OsNAC006* transcript levels were increased significantly following H_2_O_2_, NaCl, and PEG-6000 treatments, but both high and low temperature stress caused *OsNAC006* expression levels to rise then fall. Following hormone treatment, *OsNAC006* expression levels peaked at 3 and 6 h after IAA and GA_3_ treatment, respectively, while ABA treatment caused a lasting elevation in expression level (Figure 1C). The expression of *OsNAC006* varied in response to different abiotic stresses.

### 2.2. Creation of OsNAC006 Mutant Plants

The functions of NAC TFs in rice are poorly understood. We, therefore, explored the biological function of *OsNAC006* in rice. To explore the possible role of *OsNAC006* in stress responses, we generated *OsNAC006* loss-of-function lines using the CRISPR-Cas9 system. An sgRNA was designed for targeting the *OsNAC006* gene based on gene sequence information from plantTFDB (http://planttfdb_v3.cbi.pku.edu.cn). The sgRNA was cloned into the CRISPR-Cas9 T-DNA vector and transformed into plants to generate *OsNAC006*-sgRNA01 at the first exon of the *OsNAC006* gene (Figure 2A). Ten T0 lines were analyzed by enzyme digestion, and six biallelic mutations and one heterozygous mutation were identified (Figure 2B). Sanger sequencing analysis showed that the mutations included insertion of a single base pair (+1 bp/+1 bp), a single base pair deletion (−1 bp/−1 bp), and a large fragment deletion (−55 bp/−55 bp; Figure 2C). We examined the sgRNA and chose four high-probability off-target sites for the sgRNA assay for further investigations. However, we did not identify any mutations across potential off-target sites by Sanger sequencing of PCR products (Appendix A). We also screened plants that did not carry vectors in the T2 generation by further propagation and experimentation to exclude the influence of carriers. Seedlings of both *OsNAC006* mutants and WT exhibited similar growth and development dynamics under standard growing conditions. (Figure 2D).

### 2.3. OsNAC006 Mutants are Sensitive to Drought and Heat Stress

Following abiotic stress treatments, *OsNAC006* mutant expression profiles showed that growth was inhibited following drought and high temperature stress (Figure 3A,D). Further analysis revealed no differences in NBT or DAB staining between WT and *osnac006*_1 plants under standard conditions. However, after drought or heat stress, NBT and DAB staining showed that O_2_- and H_2_O_2_ levels were elevated in *osnac006* mutant plants (Figure 3B,E). The chlorophyll content was also significantly lower in mutant plants after stress treatment. Additionally, the activities of antioxidant enzymes (SOD, POD, and CAT) were decreased, and malondialdehyde (MDA), an indicator of membrane lipid peroxidation, was more abundant in *osnac006* mutant plants (Figure 3C,F). These results imply that *osnac006* may function in drought and heat tolerance by weakening the antioxidant response that is triggered to counteract oxidative stress, and by mediating photosynthesis under drought and heat stress conditions.

### 2.4. OsNAC006 Knockout Alters the Transcriptome Profile of Rice

To identify genes potentially regulated by *OsNAC006* during drought, we performed RNA-seq experiments on *osnac006*_1, *osnac006_*2, and WT plants to detect transcription profiling changes under normal and drought stress conditions. The RNA-seq results showed that gene expression was altered significantly under both stress conditions (Figure 4A,C). We selected eight genes that were significantly up- or downregulated in *osnac006* mutant plants before and after drought treatment for qRT-PCR validation of the RNA-seq results. Expression levels of all eight genes were consistent with the RNA-seq data, confirming the accuracy of the results (Appendix A).

Under standard conditions, there are 4832 genes upregulated and 1512 genes downregulated in the *osnac006_*1 mutant compared with WT plants. By comparison, 1814 genes were upregulated and 2833 genes were downregulated in the *osnac006*_2 mutant (Figure 4B). After drought stress, 527 genes were upregulated and 1209 genes were downregulated in *osnac006*_1, while 1412 genes were upregulated and 1091 were downregulated in *osnac006*_2 (Figure 4D). Six samples were tested using the BGISEQ-500 platform, with an average yield of 6.58 Gb per sample. The average alignment rate for the sample comparison genome was 88.67%, compared with 76.27% for the comparison gene set, and 570 new genes were predicted. A total of 32,482 genes were identified, including 31,922 known genes and 560 newly predicted genes. A total of 15,871 new transcripts were detected, of which 12,778 were new alternative splicing isoforms of known protein-coding genes, and 570 were transcripts of newly predicted protein-coding genes.

Venn diagram analysis revealed 1661 genes expressed in both WT and *osnac006*_1 or *osnac006*_2 mutants, which may explain the effects of knocking out *OsNAC006* on plants before treatments (Appendix A). After drought stress, the two mutants were compared with WT plants, and 793 differentially expressed genes (DEGs) were identified (Appendix A).

These 1661 and 793 DEGs identified by comparison of *osnac006*_1 and *osnac006*_2 with WT plants were subjected to Gene Ontology (GO) enrichment analysis to identify the associated biological processes (Figure 4E). DEGs related to the cellular component category were mainly associated with envelope, organelle, and intracellular organelle function. DEGs related to the molecular function category were mainly related to oxidoreductase activity, cofactor binding, and regulation terms. DEGs related to the biological process category were mainly related to oxidation-reduction process, multicellular organismal process, and response to stimulus terms. Among them, response to stimulus, organelle part, and oxidoreductase activity were the most significantly differentially expressed (Figure 4E).

### 2.5. OsNAC006 Mediates Transcriptional Responses to Drought Stress

We identified 12 enriched regions through GO analysis of DEGs altered in both *osnac006_*1 and *osnac006_*2 mutants. KEGG (Kyoto Encyclopedia of Genes and Genomes) pathway enrichment analysis was carried out to further explore the biological functions of DEGs, especially those related to membrane part, oxidoreductase activity, response to stimulus, and cofactor binding terms. The results showed that plant hormone signal transduction, MAPK signaling, diterpenoid biosynthesis, carotenoid biosynthesis, photosynthetic enzymes, photosynthesis, photosynthetic antenna proteins, ABC transporters, and starch and sucrose metabolism were among the most affected pathways (Appendix A).

We selected the most important genes of four pathways based on the KEGG results for heatmap analysis. Heatmap analysis of membrane, oxidoreductase activity, response to stimulus, and cofactor binding terms showed that DEGs belonged to various signaling pathways. Plant hormone and MAPK signaling pathways were the most significantly influenced terms related to the response to stimuli. Brassinosteroid insensitive 1 (*OsBRI1*; Os01g0718300) and 2 (*OsBIN2*; Os05g0207500), ethylene receptor *OsETR3* (Os02g0820900), auxin response factors *OsARF12* (Os04g0671900) and *OsARF19* (Os06g0702600), and ABA responsive element binding factor *OsAREB8* (Os06g0211200) are the key genes related to plant hormones (Figure 5A). Diterpenoid biosynthesis-related genes were also significantly altered. Many genes associated with diterpenoid biosynthesis including *OsGA20ox1* (Os03g0856700) can influence gibberellin-44 dioxygenase synthesis. *OsHDY1* (Os03g0685000) is an enzyme related to photosynthesis that participates in the electron transport chain and thereby influences the oxidation-reduction process (Figure 5B). *OsHPL3* (Os02g0110200), a hydroperoxide lyase, and *OsAOS1* (Os03g0767000), part of a hydroperoxide dehydratase, bind heme iron, possess monooxygenase activity, and both were significantly differentially expressed (Figure 5C). Membranes are dynamic structures that are essential for cell viability and morphogenesis. They also provide a natural interface between the environment and the cell. Diterpenoid metabolism and oxidoreductase activity related to membranes were also affected by stress treatments (Figure 5D).

## 3. Discussion

Drought stress is an important limiting factor in crop production. Approximately 20% of the world’s agricultural land is affected by drought [27]. Previous studies showed that NAC TFs are unique to plants, and not only regulate plant growth and development, but also play an important role in plant stress resistance [28,29]. Various NAC TFs in rice participate in tolerance to extreme environmental conditions. Herein, we found that the Arabidopsis TF *NAC016* promotes drought stress responses by inhibiting *AREB1* transcription. The *nac016* mutants displayed higher drought tolerance, while *NAC016* overexpressing plants (*NAC016-OX*) exhibited lower drought tolerance [30]. The *OsNAC2* overexpression line was sensitive to high salt and drought conditions. RNA interference (RNAi) can be used to increase the tolerance of plants to high salinity and drought stress [10].

In this study, we discovered that *OsNAC006* is expressed in the nucleus, and is induced by various stresses, such as abiotic and hormone stress. We used the CRISPR/Cas9 system to generate *OsNAC006* knockout mutants to characterize the role of *OsNAC006* in drought stress. *OsNAC006* mutants displayed enhanced sensitivity to drought and heat stress, which lowered chlorophyll levels, decreased POD and SOD enzyme activities, and elevated levels of MDA and other harmful oxidative damage products. Plants have evolved a complex antioxidant system involving non-enzymatic and enzymatic antioxidants [31,32]. Maintaining high levels of antioxidant enzymes such as POD, SOD, CAT, peroxidase (POX), and ascorbate peroxidase (APX) to scavenge reactive oxygen species (ROS) is associated with tolerance to stress.

Furthermore, we used RNA-seq to analyze widespread transcriptome changes under drought stress. For RNA-seq analysis of *OsNAC006* mutant plants, we focused on response to stimulus, oxidoreductase activity, cofactor binding, and membrane terms (Figure 5C). The most significant terms related to the response to stimulus subcategory were plant hormone and MAPK signal pathway genes. Hormone regulation, homeostasis, and signaling are very important in plant regulation. Some plant hormones exert strong effects on plant growth and development, such as auxins, GA, ABA, and jasmonic acid (JA), while IAA can induce growth in shoots and roots. [33]. Because plants are sessile, hormone-mediated regulation is needed to adapt to changes in the external environment [34]. Our KEGG pathway enrichment analysis revealed that many hormone biosynthetic pathways were altered. Heatmap analysis also revealed that the MAPK signaling pathway was also affected. MAPK signaling pathways are involved in the response to drought [35]. MKK3 and MPK6 were activated by JA in Arabidopsis [36], and pathogen resistance (PR) is also activated by MKK5 in response to drought stress [37]. Regarding diterpenoid biosynthesis, carbon metabolism, photosynthesis, and oxidoreductase activity were obviously affected by *OsNAC006*. Carbon metabolism is related to respiration and photosynthesis to provide energy [38].

Many binding pathways were also affected by *OsNAC006*. Heme binding, iron binding, and monooxygenase activity related to photosynthesis and respiration were altered. Previous studies showed that cells must adjust central carbon metabolism (CCM) flux via a multi-level regulatory mechanism that regulates gene expression and changes in growth conditions to rebalance the redox ratio [39].

Photosynthesis is the main driving force for plant growth, and provides the necessary energy for synthesizing organic compounds [40]. Many studies on increasing biomass production have focused on identifying genes responsible for quantitative trait loci (QTLs) to improve photosynthesis [41,42]. Membranes are essential for cell viability, morphogenesis, and maintaining normal life activities [43]. The assembly of organelles involves thousands of genes that encode a complex network of metabolic, signaling, and biosynthetic functions [44]. Heatmap results expand our understanding of the mechanisms of drought stress.

In conclusion, our transcriptomic data provide evidence that *OsNAC006* is essential for drought resistance in rice. *OsNAC006* is localized in the nucleus, and it is induced by various factors. *OsNAC006* regulates the expression of genes related to responses to stimuli, oxidoreductase activity, cofactor binding, and membrane pathways. The findings could prove valuable for genetic manipulation of drought tolerance in future plant breeding programs.

## 4. Materials and Methods

### 4.1. Plant Material and Growth Condition

The Japonica cultivar Nipponbare was employed in all transgenic experiments. RT-qPCR analysis of *OsNAC006* transcript levels was performed following eight different treatments. For RT-qPCR analysis of *OsNAC006* expression levels, we choose 4-week-old wild-type (WT) plants sown in pots and grown in a light incubator at 28 °C under a 16 h 3000 lux/8 h dark cycle. For soil drought stress treatment, evenly germinated WT and transgenic seeds were transplanted into soil and grown under normal watering conditions for 4 weeks. Drought stress was then initiated by not irrigating for 7 days. For heat stress treatment, we grown plants at 42 °C under a 16 h 3000 lux/8 h dark cycle in a light incubator.

### 4.2. OsNAC006 Expression Profile Analysis

To measure *OsNAC006* expression levels following various abiotic stress and phytohormone treatments, 4-week-old WT seedlings grown in a light incubator at 28 °C in Hoagland solution under 16 h 3000 lux/8 h dark conditions were treated with cold (4 °C), heat (42 °C), PEG 6000 (20%, *w*/*v*), NaCl (200 mm), H_2_O_2_ (1%), IAA (100 μm), ABA (100 μm) and GA_3_ (100 μm) [45]. Leaf tissue was harvested after stress treatment and subjected to RT-qPCR analysis. Three biological replicates (three independent WT plants for each abiotic stress treated sample) were examined to ensure reproducibility.

### 4.3. Subcellular Localization

In order to confirm the location of *OsNAC006*, the pZmUbi::*OsNAC006*-eGFP::HspT vector was constructed and incorporated into rice protoplasts [46,47]. The plasmid encodes *OsNAC006* fused to green fluorescent protein (GFP), and the empty GFP vector NLS::eGFP served as a control.

### 4.4. Targeted Mutagenesis of OsNAC006

We used pZHY988, the CRISPR-Cas9 backbone vector, to generate targeted *OsNAC006* mutants [48,49,50]. A single guide (sgRNA) oligonucleotide pair was designed and synthesized (Appendix A). The expression vector was transformed into *Agrobacterium tumefaciens* strain EHA105, and the resultant bacteria were used to infect rice calli [51,52]. Primers were designed and synthesized for PCR analysis (Appendix A). Amplified products were cloned into each target site, amplified by PCR, excised by restriction digestion with the corresponding enzymes, and positive clones were selected for Sanger sequencing [53,54]. All resistant callus material used to detect mutations was also used for off-target analysis. The online tool CRISPR-P (http://crispr.hzau.edu.cn/CRISPR2) was employed to predict potential off-target sites of the sgRNA, and four potential off-target sites were identified (Appendix A). We designed specific primers for further off-target analysis (Appendix A). Amplified products were cloned, and 10 positive clones were selected for Sanger sequencing.

### 4.5. Physiological Measurements

For phenotypic analysis of seedlings, WT and *OsNAC006* mutant seeds were grown to the 4-week-old seedling stage in pots, then subjected to drought or heat stress. After 7 days of treatment, physiological measurements were carried out as described in our previous study [12].

### 4.6. RNA-seq and Data Analysis

To investigate the function of *OsNAC006* in drought stress, WT, *osnac006_*1 (−55/−55) and *osnac006_*2 (T/T) plants were used for RNA-seq analysis. Five-week-old plants grown under normal conditions served as controls, and treated plants were grown for 4 weeks under normal conditions and 1 week under drought conditions. We used a mixed sample method for RNA-seq. Total RNA was extracted from mixed samples from three separate plants. For each sample, such as osnac006_1 (−55/−55) under normal condition, we selected three separate plants and pooled these into one sample, and this was complete one time. There were a total of three plants were sequenced for each of the four treatments, a total of 12 plants. RNA-seq was carried out by Beijing Genomics Institute (Shenzhen, China). Eight significantly up- and downregulated genes were selected for qRT-PCR to confirm the accuracy of the RNA-seq data. Mutant lines were assessed before and after drought treatment, and relative gene expression levels were normalized against the *Actin* gene. All assays for each gene were performed in triplicate synchronously under identical conditions. And the RNA sequences have been deposited into NCBI SRA database under accession number: PRJNA603607.

## Figures and Tables

**Figure 1 ijms-21-02288-f001:**
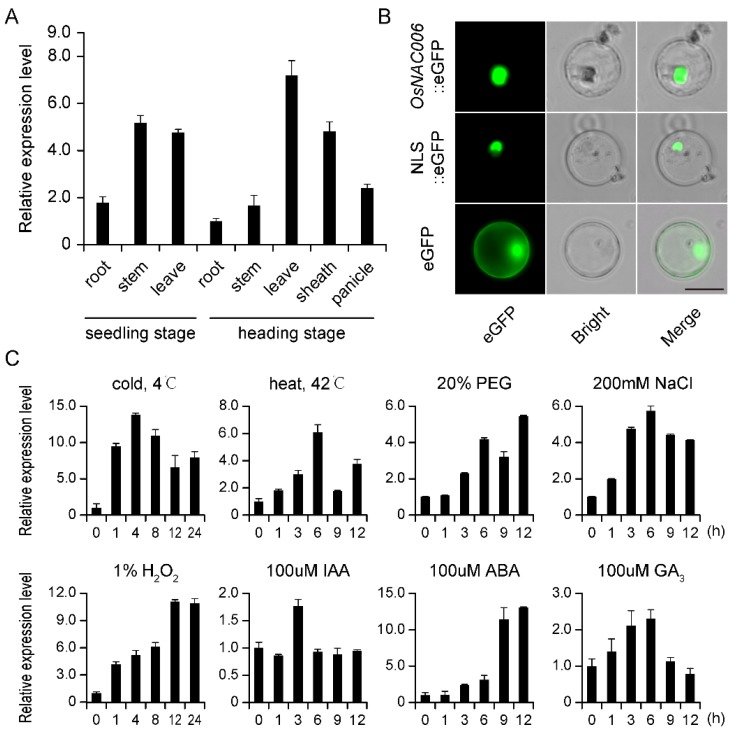
Expression profile analysis of *OsNAC006*. (**A**) Detection of *OsNAC006* expression in various tissues and organs of rice using RT-qPCR. Four-week-old seedlings were used to harvest root, sheath and leaf samples at the seedling stage. Plants in stages before the heading stage were used to harvest root, stem, sheath, leaf and panicle samples at the reproductive growth stage. Error bars indicate the standard error (SE) based on three biological replicates. (**B**), Nuclear localization of *OsNAC006* protein in the rice protoplast. NLS, Nuclear localization signal. Scale bar = 20 µm. (**C**) Expression levels of *OsNAC006* under various abiotic stresses and hormone treatments. Four-week-old seedlings were subjected to treatment with cold (4 °C), heat (42 °C), PEG 6000 (20%, *w*/*v*), NaCl (200 mm), H_2_O_2_ (1%), IAA (100 μm), ABA (100 μm) and GA3 (100 μm). The relative expression level of *OsNAC006* was measured by RT-qPCR at the indicated times. Error bars indicate SE based on three independent biological replicates.

**Figure 2 ijms-21-02288-f002:**
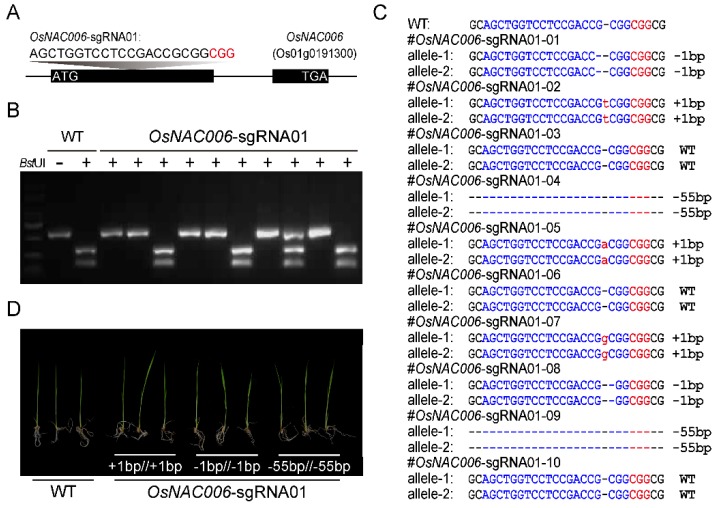
Using the CRISPR-Cas9 system to create mutants. (**A**) Design of sgRNA sites for *OsNAC006* exons. (**B**) Single-strand conformation polymorphism analysis of 11 independent *OsNAC006*-sgRNA01 T0 lines. M, Markers; WT, Wild-type. (**C**) Sanger sequencing of the target site in *OsNAC006*-sgRNA01 T0 lines. (**D**) Phenotypic analysis of *OsNAC006* T0 mutant lines.

**Figure 3 ijms-21-02288-f003:**
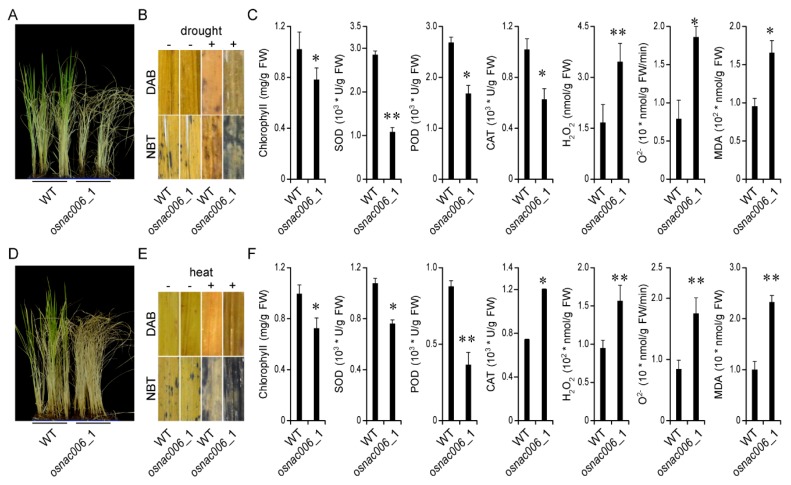
The drought-sensitive phenotype of *osnac006* mutants. (**A**) Phenotypic analysis of *OsNAC006* T1 mutant lines under drought stress. (**B**) Levels of O_2_– and H_2_O_2_ in WT and *OsNAC006* T1 mutant lines subjected to drought stress. Drought-stressed leaf samples were incubated in nitro-blue tetrazolium (NBT) or diaminobenzidine (DAB) solution. (**C**) Chlorophyll content after 20-day salt stress. Superoxide dismutase (SOD) activity after 20-day drought stress. Catalase (CAT) activity after 20-day drought stress. Peroxidase (POD) activity after 20-day drought stress. Malondialdehyde (MDA) content after 20-day drought stress. H_2_O_2_ content after 20-day drought stress. O_2_– production rate after 20-day drought stress. Bars represent the mean ± SE of three independent experiments. (**D**) Phenotypic analysis of *OsNAC006* T1 mutant lines under heat stress. (**E**) Levels of O_2_– and H_2_O_2_ in WT and *OsNAC006* T1 mutant lines subjected to heat stress. Heat-stressed leaf samples were incubated in nitro-blue tetrazolium (NBT) or diaminobenzidine (DAB) solution. (**F**) Chlorophyll content after 4-day heat stress. Superoxide dismutase (SOD) activity after 4-day heat stress. Catalase (CAT) activity after 20-day salt stress. Peroxidase (POD) activity after 4-day heat stress. Malondialdehyde (MDA) content after 20-day drought stress. H_2_O_2_ content after 4-day heat stress. O_2_– production rate after 4-day heat stress. Bars represent the mean ± SE of three independent experiments. ∗ and ∗ ∗ represent significant differences at *p* < 0.05 and *p* < 0.01 compared to WT.

**Figure 4 ijms-21-02288-f004:**
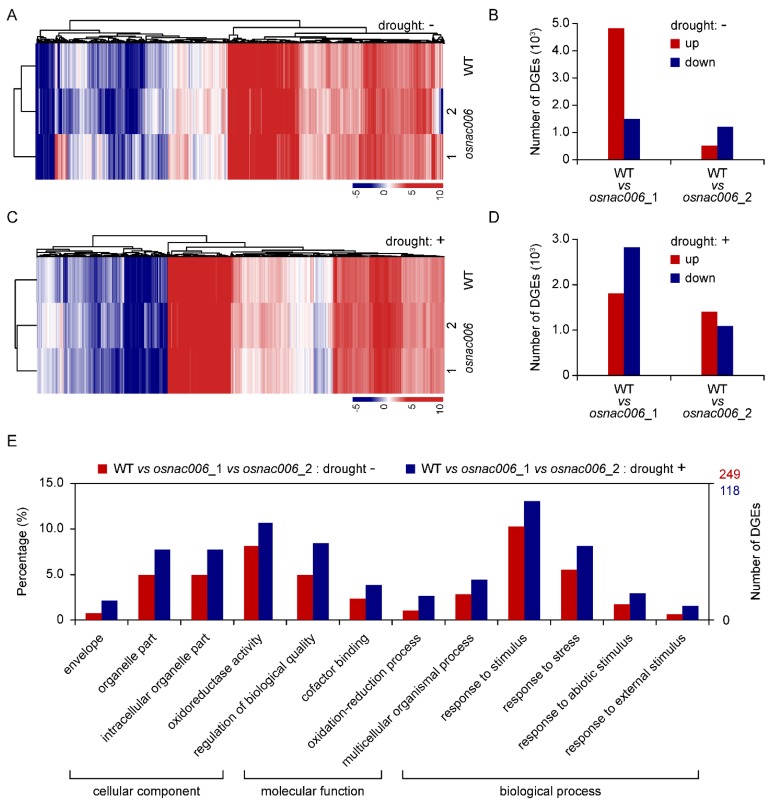
Global gene expression changes in knockout *OsNAC006* rice. (**A**) The most significant clustering analysis of differentially expressed genes (DEGs) in WT and *osnac006* T1 mutant lines. Targeted knockout of *osnac006* resulted in profound changes to gene expression, physiology, and development compared with WT and controls without drought stress treatment. The colour scale corresponds to log2 (FPKM) values of the genes. (**B**) Number of DEGs in WT, *osnac006*_1 and *osnac006*_2 T1 mutant lines, based on expression profiles obtained by RNA-Seq. Total RNA was extracted from mixed samples from three separate plants. (**C**) DEGs shared by WT and *osnac006*_1 and WT and *osnac006*_2 lines before drought stress. (**D**) DEGs shared by WT and *osnac006*_1 and WT and *osnac006*_2 lines after drought stress. (**E**) Gene ontology (GO) classification of DEGs shared by WT and *osnac006*_1 and WT and *osnac006*_2 mutant lines under normal and drought stress conditions. The x-axis shows user-selected GO terms, and the y-axis shows the percentage of genes (number of a particular gene divided by the number of total genes).

**Figure 5 ijms-21-02288-f005:**
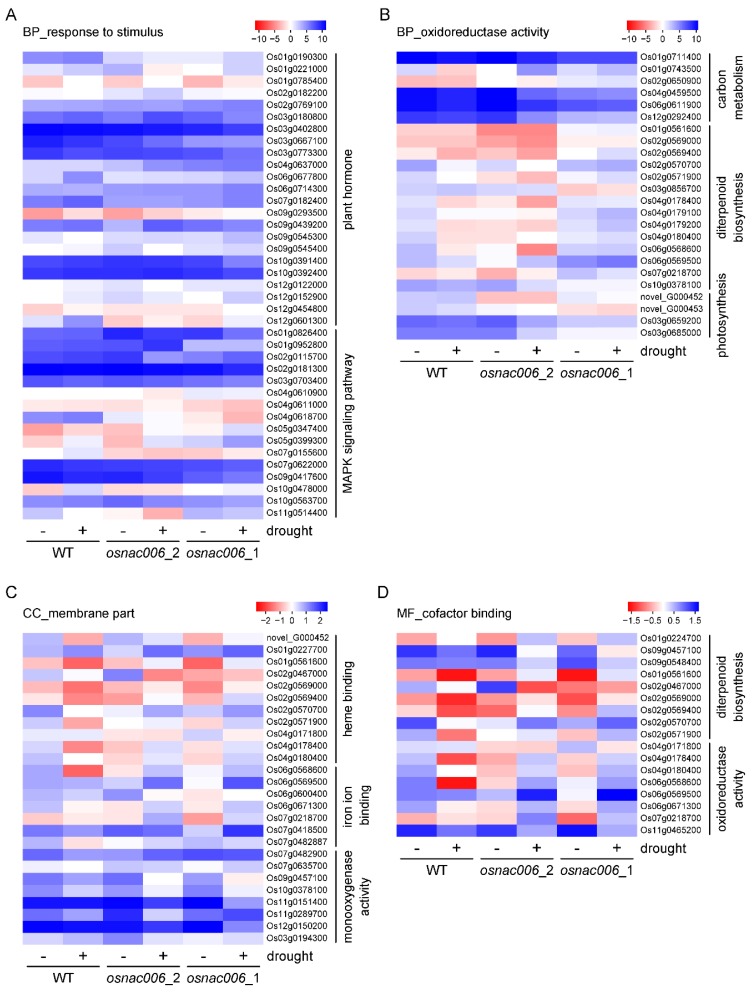
Transcriptome analysis of genes systemically regulated in WT and *osnac006* T1 mutant lines in response to drought stress. (**A**) Response to stimuli. (**B**) Oxidoreductase activity. (**C**) Membrane part. (**D**) Cofactor binding. Log_2_ fold change (FC) values for DEGs in WT and *osnac006**_*1 and *osnac006_*2 mutant lines are shown before (drought−) and after (drought+) drought treatment.

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
