# Peer review of "Knockout of the *OsNAC006* Transcription Factor Causes Drought and Heat Sensitivity in Rice"

_ijms, 2020, doi:10.3390/ijms21072288_

Round 1
Reviewer 1 Report
Dear authors
I have read your work with great interest and I compliment the results described. However, there are a number of criticisms that I hope will be useful for the publication of the work.
First of all, I believe that the introduction can be enriched: for example, I would introduce some sentences to describe the importance of using the CRISPR technique in the genetic transformation of plants of agricultural interest. In fact, in many countries these techniques are still considered GM and, as such, are prohibited. Therefore, I strongly advise the authors to specify the advantages of this technique compared to classical transgenosis, also in the light of the political and social problems that are still encountered especially in Europe (see, for example, Rugini, E., Silvestri, C., Cristofori, V., Brunori, E., & Biasi, R. (2015). Ten years field trial observations of ri-TDNA cherry colt rootstocks and their effect on grafted sweet cherry cv lapins. Plant Cell, Tissue and Organ Culture, 123(3), 557-568. doi:10.1007/s11240-015-0860-x).
As for the English language, I think the work is very well written (but I am not a native English speaker). However, there are errors in the text that need to be corrected. Below are some errors that I have noticed:
Line 24: treatments
Line 27: "involved responses to stimuli" change with "involved in response to stimuli"
Line 28: Illuminate, change with elucidate
Keywords: the keywords must be different from the word used in the title. So, I suggest to remove OsNAC006 and, furthermore, I suggest to delete drought and heat, and replace with "abiotic stresses"
Line 52: "In our present work" replace with "In this work"
Line 55: osnac006: please write it in the correct form
Line 64: It is not necessary to specified "Quantitative real timepolymerase chain reaction" because this is a well-known techniques for the readers.
Line 75: How the authors decide the concentrations of the simulated stress imposed? I think they refer to other literature, so please justify how the authors used these concentrations
Line 103: treatment plants, replace with treated plants
Line 108: conditions
Line 117: lack of parenthesis
A major concern of the work is the lack of a statistical analysis. All the paper need a statistical analisys to justify the results reported
Author Response
Comments to the Author
1.Dear authors
I have read your work with great interest and I compliment the results described. However, there are a number of criticisms that I hope will be useful for the publication of the work.
First of all, I believe that the introduction can be enriched: for example, I would introduce some sentences to describe the importance of using the CRISPR technique in the genetic transformation of plants of agricultural interest. In fact, in many countries these techniques are still considered GM and, as such, are prohibited. Therefore, I strongly advise the authors to specify the advantages of this technique compared to classical transgenosis, also in the light of the political and social problems that are still encountered especially in Europe (see, for example, Rugini, E., Silvestri, C., Cristofori, V., Brunori, E., & Biasi, R. (2015). Ten years field trial observations of ri-TDNA cherry colt rootstocks and their effect on grafted sweet cherry cv lapins. Plant Cell, Tissue and Organ Culture, 123(3), 557-568. doi:10.1007/s11240-015-0860-x).
Authors’ response: We are very grateful for the positive comments that the reviewer regarding our work. As the reviewer rightfully pointed out, we realize that adding the difference between traditional techniques and CRISPR/Cas9 gene editing techniques will make the article richer and more complete. We are particularly grateful to the editor for the references provided to us, which are of great help to us in polishing the article. We added a paragraph in the introduction: “CRISPR/Cas9 gene editing technology is gradually applied to many genes related to rice breeding, which is of great significance for agricultural breeding [25]. The traditional transgenic technology is based on T-DNA insertion technology, and the vector transferred into the plant will not disappear [26]. CRISPR/Cas9-mediated genome editing has attracted people's attention not only because of its simplicity, accuracy and efficiency, but also because of its ability to produce non-transgenic plants [27]. The mutant plants that had produced the required mutations can lost the CRISPR/Cas9 vectors through several generations of character isolation. With the emergence of CRISPR/Cas9 gene editing technology, it is convenient for us to understand the gene function of plants. The new generation of breeding technology based on CRISPR/Cas9 editing system is gradually maturing.” We have cited the paper (Rugini, E., Silvestri, C., Cristofori, V., Brunori, E., & Biasi, R. (2015). Ten years field trial observations of ri-TDNA cherry colt rootstocks and their effect on grafted sweet cherry cv lapins. Plant Cell, Tissue and Organ Culture, 123(3), 557-568. doi:10.1007/s11240-015-0860-x ) in the revised manuscript.
- Line 24: treatments
Authors’ response: As for the English language, we are very sorry to have some English spelling problems, we are very grateful to the editor for his comments, and we have seriously corrected the mistakes.
Sorry about the error. It has been fixed at line 25.
- Line 27: "involved responses to stimuli" change with "involved in response to stimuli".
Authors’ response: Sorry about the error, we have made the change at line 29.
- Line 28: Illuminate, change with elucidate
Authors’ response: Thanks for the suggestion. “Illuminate” had changed to “elucidate” at line 31.
- Keywords: the keywords must be different from the word used in the title. So, I suggest to remove OsNAC006 and, furthermore, I suggest to delete drought and heat, and replace with "abiotic stresses".
Authors’ response: We agree with this opinion on key words very much. We had deleted OsNAC006, drought and heat. We add “abiotic stresses” to keywords at line [33-34].
- Line 52: "In our present work" replace with "In this work"
Authors’ response: "In our present work" had changed to "In this work". And this paragraph has been modified by the suggestion of another editor at line 74.
- Line 55: osnac006: please write it in the correct form
Authors’ response: Sorry about the error, this part has been deleted
- Line 64: It is not necessary to specified "Quantitative real timepolymerase chain reaction" because this is a well-known techniques for the readers.
Authors’ response: Thanks for the suggestion. This part has been deleted at line 80.
- Line 75: How the authors decide the concentrations of the simulated stress imposed? I think they refer to other literature, so please justify how the authors used these concentrations
Authors’ response: We thank the reviewer for raising this point. As the reviewer said, we did refer to the relevant literature to determine the concentration of the treatment. We have added the reference, Xiong, H., et al., Natural Variation in OsLG3 Increases Drought Tolerance in Rice by Inducing ROS Scavenging. Plant Physiol, 2018. 178(1): p. 451-467. This document can provide us with a reference to the concentration of hormones and other treatments. At the same time, we carried out pre-experiments in the early stage to find a more appropriate treatment concentration. Finally, combination of literature and pre-experiment, the treatment concentration in this experiment was determined.
- Line 103: treatment plants, replace with treated plants.
Authors’ response: Thanks for the suggestion. We have made the change at line 120.
- Line 108: conditions
Authors’ response: Sorry about the error. “for each of the 4 consition, a total of 12 of plants ” has been changed to “for each of the four treatments, a total of 12 plants” at line 125.
- Line 117: lack of parenthesis
Authors’ response: Sorry about the error. It has been fixed at line 135.
- A major concern of the work is the lack of a statistical analysis. All the paper need a statistical analisys to justify the results reported
Authors’ response: Thanks for the suggestion. Your opinion is very important. We added a statistical analysis in the paper and made a significant analysis of the data of the article. It has been added to our results through calculation in the Figure 3. We have updated Figure 3. Here, ∗ and ∗ ∗ represent significant differences at p < 0.05 and p < 0.01 compared to WT.

Reviewer 2 Report
The authors should consider the following suggestions to improve the manuscript for publication:
Line 21) “There are 170 NAC TFs in rice,…” suggest change to “To date, 170 NAC TFs have been reported in rice,…”
Line 22) “Herein, we discovered that OsNAC006 is constitutively expressed,…” suggest change to “Herein, we discovered that the TF OsNAC006 is constitutively expressed in rice,…”
Line 25) “…CRISPR-Cas9 system showed that loss of OsNAC006 conferred drought and heat sensitivity.” Suggest change to “…CRISPR-Cas9 system resulted in drought and heat sensitivity.”
Lines 35-36) “…different abiotic stresses can affect plant growth [1, 2].” Suggest change to “…different abiotic stresses can affect plant growth and crop performance [1, 2].”
Line 37) “…and it causes…” suggest change to “…and causes…”
Line 40) “…including regulating transcription factors (TFs).” Are there recent studies that specifically support this statement that could be cited?
Line 41) “In rice, 2408 TFs have been identified and classified into 56 families.” In the abstract, “170 NAC TF” is stated but not substantiated. Perhaps that assertion should be restated here with a citation?
Lines 52-60) This paragraph presents information that is more appropriate for the results section. The reviewer suggests a change to the following: “In our present work, we cloned the rice NAC TF-encoding gene OsNAC006 (LOC_Os01g09550) and present evidence that mutations of this gene confer drought and heat sensitivity.”
Line 63) “The O. sativa L. japonica cultivar…”. “japonica” is a subspecies name, not a cultivar name. What was the name of the specific cultivar that was used? There are thousands of japonica cultivars.
Line 73) “…in Hoagland solution…”. There are several variants of this nutrient solution; a citation to the formulation should be provided.
Lines 107-108) “…for each of the 4 consition, a total of 12 of plants.” Change to “…for each of the four treatments, a total of 12 plants.”
Line 117) “…and panicle from the heading stage.” Add closed parenthesis “…and panicle from the heading stage).”
Lines 130-131) “In general, the expression of OsNAC006 varied was in response to multiple stresses.” Change to “The expression of OsNAC006 varied in response to different abiotic stresses.”
Line 150) “…ad six biallelic mutations…” change to “…and six biallelic mutations…”
Lines 152-153) “…single base pair deletion (-1 bp / -1 bp), and large fragment deletion…” change to “…a single base pair deletion (-1 bp / -1 bp), and a large fragment deletion…”
Line 154) “…to investigate potential off-targets.” Change to “…for further investigations.”
Lines 157-158) “OsNAC006 mutant seedlings grew almost as well as WT plants under standard growth temperature conditions (Fig. 2D).” Suggest change to “Seedlings of both OsNAC006 mutants and WT exhibited similar growth and development dynamics under standard growing conditions (Fig. 2D).”
Lines 164-165) “Under normal growth temperature conditions, there were no differences between WT and mutant plants, but after stress treatment, OsNAC006 mutant expression profiles showed…”. This is redundant; change to “Following abiotic stress treatments, OsNAC006 mutant expression profiles showed…”
Line 168) “…normal..” suggest that the authors use “…standard…” consistently.
Line 200) “…osnac006…” should it be italicized?
Line 203) “…normal…” à “…standard…”?
Line 221) “…terms.” Suggest change to “…function.”
Line 230) “…osnac006 resulted in global changes…” suggest change to “…osnac006 resulted in profound changes to gene expression, physiology, and development…”
Line 255) “…enzymes related to plant hormones..” Is “enzymes” the proper term? It seems like it should be “genes”.
Line 289) “…global transcriptome…” suggest change to “…widespread transcriptome…”
Line 295) “…static,…” the reviewer suggests change to “…sessile,…”
Line 304) “Many binding pathways were also affected.” Suggest change to “Many binding pathways were also affected by OsMAC006.”
Line 308) “…can provide…” suggest change to “…provides…”
Line 313) “Our heatmap…” suggest change to “Heatmap…”
Author Response
- Line 21) “There are 170 NAC TFs in rice,…” suggest change to “To date, 170 NAC TFs have been reported in rice,…”
Authors’ response: We thank the reviewer for pointing out the expression is not accurate enough. “There are 170 NAC TFs in rice,…” has been changed to “To date, 170 NAC TFs have been reported in rice,…” at line 21.
- Line 22) “Herein, we discovered that OsNAC006 is constitutively expressed,…” suggest change to “Herein, we discovered that the TF OsNAC006 is constitutively expressed in rice,…”
Authors’ response: Thanks for the suggestion. We have made the change at line 22.
- Line 25) “…CRISPR-Cas9 system showed that loss of OsNAC006 conferred drought and heat sensitivity.” Suggest change to “…CRISPR-Cas9 system resulted in drought and heat sensitivity.”
Authors’ response: Thanks for the suggestion. We have made the change at line 26.
- Lines 35-36) “…different abiotic stresses can affect plant growth [1, 2].” Suggest change to “…different abiotic stresses can affect plant growth and crop performance [1, 2].”
Authors’ response: Thanks for the suggestion. We have made the change at line 37-38.
- Line 37) “…and it causes…” suggest change to “…and causes…”
Authors’ response: Sorry about the error, it has been fixed at line 40.
- Line 40) “…including regulating transcription factors (TFs).” Are there recent studies that specifically support this statement that could be cited?
Authors’ response: Thanks for the suggestions. We have updated the references [10-12] at line 43.
- Line 41) “In rice, 2408 TFs have been identified and classified into 56 families.” In the abstract, “170 NAC TF” is stated but not substantiated. Perhaps that assertion should be restated here with a citation?
Authors’ response: Thanks for the suggestions. We have added data sources at line [44-45], “data provided by Plant Transcription Factor Database v3.0, Center for Bioinformatics, Peking University.”
- Lines 52-60) This paragraph presents information that is more appropriate for the results section. The reviewer suggests a change to the following: “In our present work, we cloned the rice NAC TF-encoding gene OsNAC006 (LOC_Os01g09550) and present evidence that mutations of this gene confer drought and heat sensitivity.”
Authors’ response: Thanks for the suggestions. We think the editor's suggestion makes the expression of the paragraph more appropriate. So we deleted this paragraph, and changed to “In this work, we cloned the rice NAC TF-encoding gene OsNAC006 (LOC_Os01g09550) and present evidence that mutations of this gene confer drought and heat sensitivity.”
- Line 63) “The O. sativa L. japonica cultivar…”. “japonica” is a subspecies name, not a cultivar name. What was the name of the specific cultivar that was used? There are thousands of japonica cultivars.
Authors’ response: Thanks for the suggestion. We have made the change. “The O. sativa L. japonica cultivar…” has been changed to “The Japonica cultivar Nipponbare”
- Line 73) “…in Hoagland solution…”. There are several variants of this nutrient solution; a citation to the formulation should be provided.
Authors’ response: Thanks for the suggestions. We have updated the reference [28].
- Lines 107-108) “…for each of the 4 consition, a total of 12 of plants.” Change to “…for each of the four treatments, a total of 12 plants.”
Authors’ response: Thanks for the suggestions. “…for each of the 4 consition, a total of 12 of plants.” has been changed to “…for each of the four treatments, a total of 12 plants.” at line [124-125].
- Line 117) “…and panicle from the heading stage.” Add closed parenthesis “…and panicle from the heading stage).”
Authors’ response: Sorry about the error. It has been fixed at line 135.
- Lines 130-131) “In general, the expression of OsNAC006 varied was in response to multiple stresses.” Change to “The expression of OsNAC006 varied in response to different abiotic stresses.”
Authors’ response: Thanks for the suggestions. “In general, the expression of OsNAC006 varied was in response to multiple stresses.” has been changed to “The expression of OsNAC006 varied in response to different abiotic stresses.” at line [148-149]
- Line 150) “…ad six biallelic mutations…” change to “…and six biallelic mutations…”
Authors’ response: Sorry about the error. It has been fixed at line 168.
- Lines 152-153) “…single base pair deletion (-1 bp / -1 bp), and large fragment deletion…” change to “…a single base pair deletion (-1 bp / -1 bp), and a large fragment deletion…”
Authors’ response: Sorry about the error. It has been fixed at line [170-171].
- Line 154) “…to investigate potential off-targets.” Change to “…for further investigations.”
Authors’ response: Thanks for the suggestions. “…to investigate potential off-targets.” has been changed to “…for further investigations.” at line 173
- Lines 157-158) “OsNAC006 mutant seedlings grew almost as well as WT plants under standard growth temperature conditions (Fig. 2D).” Suggest change to “Seedlings of both OsNAC006 mutants and WT exhibited similar growth and development dynamics under standard growing conditions (Fig. 2D).”
Authors’ response: Thanks for the suggestions. It has been fixed at line [176-179].
- Lines 164-165) “Under normal growth temperature conditions, there were no differences between WT and mutant plants, but after stress treatment, OsNAC006 mutant expression profiles showed…”. This is redundant; change to “Following abiotic stress treatments, OsNAC006 mutant expression profiles showed…”
Authors’ response: Thanks for the suggestions. It has been changed at line [186-187].
- Line 168) “…normal..” suggest that the authors use “…standard…” consistently.
Authors’ response: Thanks for the suggestions. It has been changed at line 190.
- Line 200) “…osnac006…” should it be italicized?
Authors’ response: Sorry about the error. It has been fixed at line 223.
- Line 203) “…normal…” à “…standard…”?
Authors’ response: Thanks for the suggestions. It has been changed at line 226.
- Line 221) “…terms.” Suggest change to “…function.”
Authors’ response: Thanks for the suggestions. It has been changed at line 244.
- Line 230) “…osnac006 resulted in global changes…” suggest change to “…osnac006 resulted in profound changes to gene expression, physiology, and development…”
Authors’ response: Thanks for the suggestions. It has been changed at line [253-254].
- Line 255) “…enzymes related to plant hormones..” Is “enzymes” the proper term? It seems like it should be “genes”.
Authors’ response: Thanks for the suggestions. “enzymes” has been changed to “genes” at line 279.
- Line 289) “…global transcriptome…” suggest change to “…widespread transcriptome…”
Authors’ response: Thanks for the suggestions. “…global transcriptome…” has been changed to “…widespread transcriptome…” at line 313.
- Line 295) “…static,…” the reviewer suggests change to “…sessile,…”
Authors’ response: Thanks for the suggestions. “…static,…” has been changed to “…sessile,…” at line 320.
- Line 304) “Many binding pathways were also affected.” Suggest change to “Many binding pathways were also affected by OsMAC006.”
Authors’ response: Thanks for the suggestions. It has been changed at line 328.
- Line 308) “…can provide…” suggest change to “…provides…”
Authors’ response: Thanks for the suggestions. It has been changed at line 333.
- Line 313) “Our heatmap…” suggest change to “Heatmap…”
Authors’ response: Thanks for the suggestions. It has been fixed at line 338.

Round 2
Reviewer 1 Report
Dear Authors,
I think you have address all the criticisms observed.
Best regards